palaeontology/biomechanics

echinoderm, Stylophora, aulacophore, three-dimensional digital modelling, range of motion

**Author for correspondence:**
Elizabeth G. Clark
e-mail: elizabeth.g.clark@yale.edu

# Arm waving in stylophoran echinoderms: three-dimensional mobility analysis illuminates cornute locomotion

Elizabeth G. Clark[1], John R. Hutchinson[2], Peter J. Bishop[2,3] and Derek E. G. Briggs[1,4]

[1]Department of Geology and Geophysics, Yale University, New Haven, CT 06511, USA
[2]Structure and Motion Laboratory, Department of Comparative Biomedical Sciences, The Royal Veterinary College, Hatfield AL9 7TA, UK
[3]Geosciences Program, Queensland Museum, Brisbane, Australia
[4]Yale Peabody Museum of Natural History, Yale University, New Haven, CT 06511, USA

EGC, 0000-0003-4289-6370; JRH, 0000-0002-6767-7038;
PJB, 0000-0003-2702-0557; DEGB, 0000-0003-0649-6417

The locomotion strategies of fossil invertebrates are typically interpreted on the basis of morphological descriptions. However, it has been shown that homologous structures with disparate morphologies in extant invertebrates do not necessarily correlate with differences in their locomotory capability. Here, we present a new methodology for analysing locomotion in fossil invertebrates with a rigid skeleton through an investigation of a cornute stylophoran, an extinct fossil echinoderm with enigmatic morphology that has made its mode of locomotion difficult to reconstruct. We determined the range of motion of a stylophoran arm based on digitized three-dimensional morphology of an early Ordovician form, *Phyllocystis crassimarginata*. Our analysis showed that efficient arm-forward epifaunal locomotion based on dorsoventral movements, as previously hypothesized for cornute stylophorans, was not possible for this taxon; locomotion driven primarily by lateral movement of the proximal aulacophore was more likely. Three-dimensional digital modelling provides an objective and rigorous methodology for illuminating the movement capabilities and locomotion strategies of fossil invertebrates.

## 1. Introduction

Living echinoderms exhibit pentameral symmetry, allowing them to move omnidirectionally [1]. Each extant class has developed innovative strategies for locomotion. Early fossil echinoderms, in contrast, demonstrate a range of symmetries: spiral-radiate,

pentameral, bilateral and asymmetric forms were all present by the Cambrian Wuliuan Age [2]. The evolution of diverse symmetries from a bilateral ancestor represents one of the most radical body plan divergences in the history of complex animals, with interesting consequences for the evolution of locomotion within the phylum. The locomotory strategies of many early echinoderm groups remain imperfectly understood. This includes stylophorans, which ranged from Cambrian (Wuliuan) to Carboniferous (Bashkirian; [2,3]), and have been described as some of 'the strangest-looking animals that have ever existed' [4]. Their body plan varied from weakly to strongly asymmetrical [5] and their ability to perform locomotion is debated [5–9].

Stylophorans had a plated body (theca) and a segmented appendage (aulacophore). Some interpretations considered stylophorans to be mostly sessile [5,10–11]. Aspects of their morphology, however, including the apparent flexibility of the aulacophore in certain taxa, the lack of a holdfast, and evidence for the presence of musculature in the proximal aulacophore [12] suggest that they may have been capable of locomotion. Trackways have also been found associated with stylophoran body fossils [8].

Mechanics have been posited for three strategies for aulacophore-powered locomotion in cornute stylophorans [13,14]. These hypothesized strategies suggest that movement was aulacophore-forward and epifaunal, as the shape of the theca and the curvature of certain features would have facilitated motion in that direction [13,14]. The three strategies differ primarily in the movement of the aulacophore used to drive locomotion. In the first, locomotion is driven by dorsoventral oscillations of the aulacophore [13]. For this strategy to be effective, the aulacophore would need to be extended and inserted into the substrate at a relatively shallow angle, swept through the sediment, and removed at a relatively steep angle to the plane of the body [13: fig. 6]. The second strategy involves locomotion driven by lateral movements of the aulacophore. A lateral/ventral power stroke was followed by a dorsal/lateral return stroke [14], both driven by a muscular distal aulacophore [14]. In the third strategy, a muscular proximal aulacophore would have driven locomotion [14]. This inference is based on the results of simulations performed using a wooden model with a centre of rotation where a rigid aulacophore inserts into the theca [14].

The locomotory capabilities of stylophorans have typically been reconstructed by assigning functional attributes to specific morphological features [5–7,13–15]. However, evidence suggests that disparate morphological features in living echinoderms do not necessarily reflect a difference in function [16]: functional capabilities cannot be determined based on qualitative aspects alone. Biomechanical modelling of digitized fossil specimens has been used to calculate range of motion and analyse locomotion in extinct vertebrates [17–18]. Three-dimensional digital modelling has also been used to analyse invertebrate biomechanics, including the range of motion or movement in living cockroaches [19] and brittle stars [16]. Our investigation of stylophorans, to our knowledge, represents the first instance in which similar methods have been applied to estimate range of motion for a fossil invertebrate.

A three-dimensional digital model based on a micro-computed tomographic (CT) scan of *Phyllocystis crassimarginata* (Stylophora, Cornuta; [20]) from the Lower Ordovician of France allowed us to reconstruct the range of motion of the aulacophore. Constraining the range of motion of the aulacophore in this way is the first step towards determining how locomotion was achieved in stylophorans and evaluating the feasibility of several hypothesized locomotion strategies.

## 2. Material and methods

Our investigation was based on a specimen of *Phyllocystis crassimarginata* [20] (UCBL-FSL 712515, Université Claude Bernard Lyon 1, Villeurbanne) from the Upper Tremadocian, Saint-Chinian Formation, *Euloma filacovi* Zone, northwest of Felines-Minervois, Merlaux Valley, France [21], which is preserved as a mould within a concretion (figure 1). The preservation is almost entirely three-dimensional, the near-complete morphology of the proximal and distal aulacophore is evident, and the ossicles of the aulacophore largely retain the articulations as they were presumably in life.

The two halves of the concretion were reassembled and scanned using an X-Tek H 225 µCT scanner (Nikon Metrology, Tring, UK) at the University of Cambridge Museum of Zoology (1080 projections at 155 kV and 201 µA for a voxel size of 19.5 µm). The preservation facilitated scanning: the selected area was inverted so that the volume of the air (i.e. the mould) was reconstructed. Surface structures of the endoskeleton were extracted as three-dimensional polygonal meshes from VG Studio MAX 3.0 (Volume Graphics, Heidelberg, Germany) and imported into Maya software (Autodesk, San Rafael, USA) (see [16]). Micro-CT scans and Maya models are available through the Dryad Digital Repository [22].

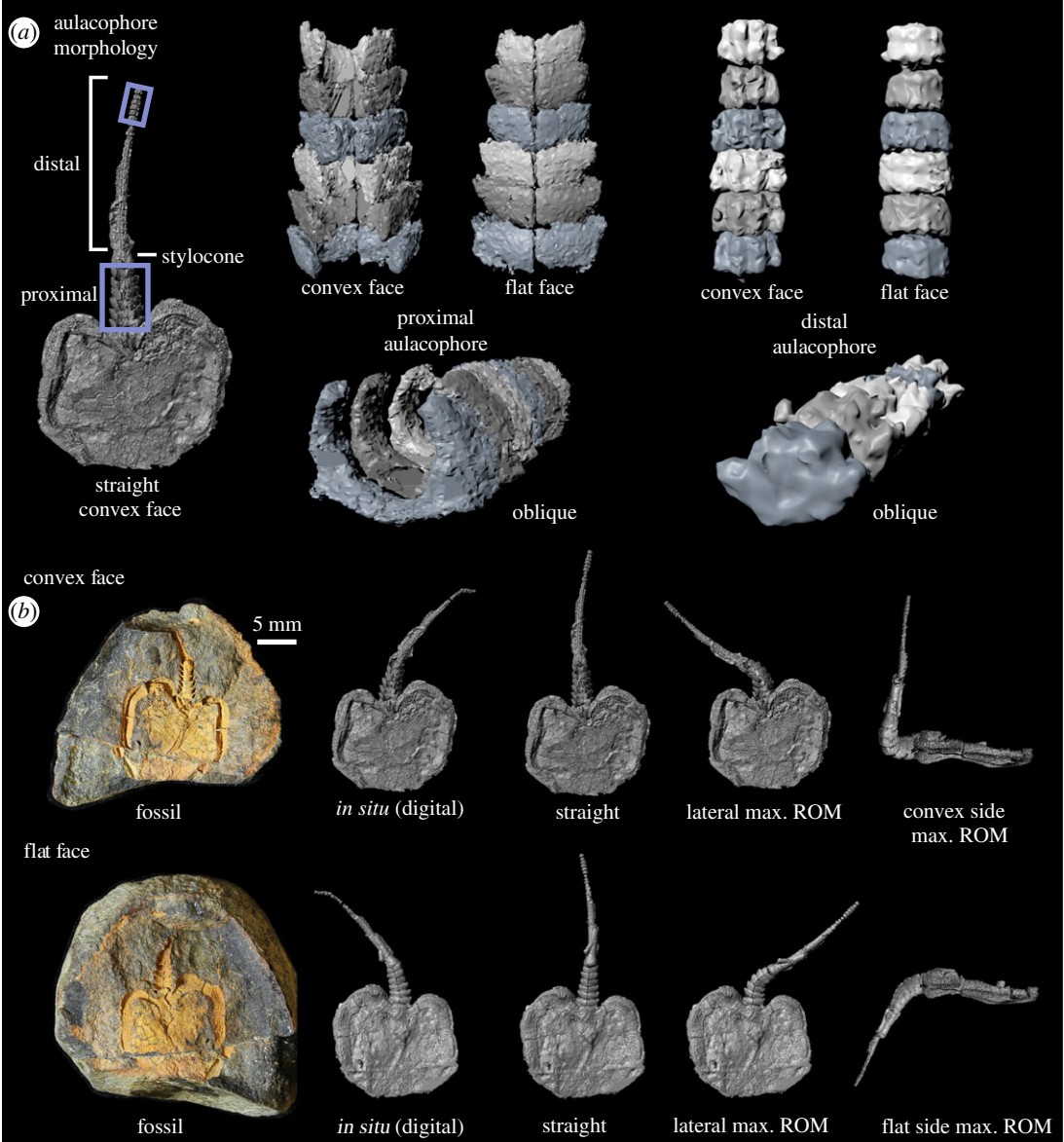

**Figure 1.** Stylophoran *Phyllocystis crassimarginata*. (*a*) Morphology of the aulacophore. The segments of the proximal aulacophore consist of two tectals (not illustrated) and two inferolaterals, creating a hollow tube [5]. The segments of the distal aulacophore are composed of two cover plates (not preserved here) and one larger ossicle. The stylocone is a massive ossicle connecting the proximal and distal portions of the aulacophore. (*b*) *In situ* position (left; both faces of fossil) and estimated maximal (max.) ranges of motion (ROM) versus *in situ* and straightened poses (right).

The surface structure of the theca was exported as a single mesh. Twelve inferolateral ossicles from the proximal aulacophore were extracted as individual meshes. The tectals were not articulated *in situ* and their position was not reconstructed. The stylocone and the proximal part of the distal aulacophore were not readily differentiated and were extracted as a single mesh (the flat proximal and distal surfaces and small spaces between serial segments suggest that they operated as essentially semi-rigid beams). The nine distalmost aulacophore segments were slightly disarticulated; they were extracted as individual meshes and digitally reassembled assuming articulations similar to those in the proximal part of the distal aulacophore.

The two paired inferolateral ossicles in each segment were aligned so that they were near-symmetrical and rotated and translated to align those of successive segments. Segments were articulated with spacing equivalent to their position *in situ* and assembled hierarchically into a digital marionette [16,23]: movement of proximal ossicles resulted in movement of those downstream. Joint centres were estimated by fitting geometric primitives to the meshes of the inferolateral pairs using a custom script

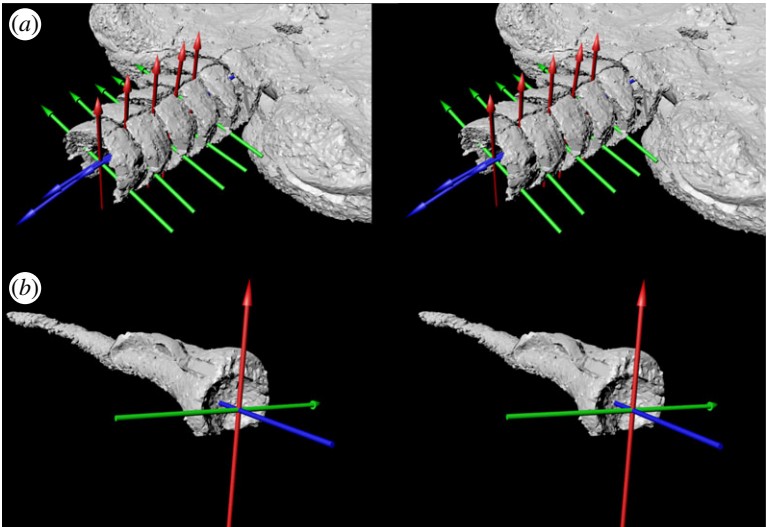

**Figure 2.** Stereo images of the proximal aulacophore (*a*) and stylocone (*b*) of *Phyllocystis crassimarginata* (UCBL-FSL 712515). Tricoloured axes represent the coordinate system of the joint centre (see Material and methods). The centre of each axis is positioned at the joint centre and demarcates three rotational degrees of freedom: dorsoventral (red, *x*-axis), mediolateral (green, *y*-axis) and internal/external (blue, *z*-axis). Orientation of the coordinate system is based on [16]. Visualized using Autodesk Maya.

in Matlab software (The MathWorks, Inc., Natick, MA, USA) which optimized the match of a sphere to the polygonal mesh. This procedure established a mediolateral joint axis; the proximodistal axis of the joint was aligned with the axis of the arm and the dorsoventral axis lay orthogonal to these. The centre of the joint was at the intersection of all three axes, on the midline of the inferolateral pairs and the proximal surface of the stylocone (figure 2).

To estimate the maximal range of motion at each inter-ossicle joint of the proximal aulacophore, each pair of inferolaterals was rotated at the joint centre until they collided with the edge of the adjacent proximal inferolateral. Range of motion was reconstructed in dorsoventral and lateral planes (assuming left–right aulacophore symmetry).

# 3. Results

## 3.1. Morphology of *Phyllocystis crassimarginata* (figure 1)

The theca is subcircular in outline with an invagination where the aulacophore inserted. The posterior margin [9: fig. 1*a*] echinoderm interpretation) of the theca is incomplete where it extends beyond the edge of the concretion. The marginals which frame the theca extend slightly beyond the body cavity on the convex surface [5]; the opposing surface is relatively flat.

The proximal aulacophore is composed of six segments, each with a pair of inferolateral ossicles which are roughly C-shaped in distal view. Opposing members of a pair articulate across the aulacophore midline and do not appear to have changed position relative to one another during aulacophore flexion. The width of each inferolateral increases distally along its length to accommodate the succeeding segment, forming a series of telescoping rings. Successive inferolaterals decrease slightly in width. The hollow tube created by the proximal aulacophore connects with the internal body cavity.

The stylocone, a single ossicle connecting the proximal and distal aulacophore, tapers distally. The proximal surface has a rounded depression. There is a small groove running along the anterior–posterior axis on the surface corresponding to the convex side of the theca.

The distal aulacophore is characterized by a continuation of the groove, flanked by two ridges. Lateral projections have been interpreted as movable cover plates [9]. The segments are wider than long and separated by relatively flat articulations orthogonal to the axis of the aulacophore. The aulacophore is slightly convex on the side corresponding to the flat surface of the theca.

**Table 1.** Intersegmental range of motion. Angles with values in degrees (theta) created by successive segments in the range of motion models. As the proximalmost segment is fused to the theca, the 'axis' between the theca and this first segment has the same orientation as the axis of the second segment in the straightened pose. The angles between this imaginary first axis and the actual first one in the model are reported in the 1 → 2 row. The rotations about the 'primary axes' (lateral: Y; flat and convex side: Z) contribute to the cumulative total angular displacement from base to tip in the primary direction of motion. In several cases with relatively small angles (e.g. lateral 5 → 6 and flat side 5 → 6), the different sign is a consequence of the interacting effects of rotations about the other two joint axes. This may be due to the non-uniform orientation of the axes in the straightened reference pose.

| aulacophore flexion | segment angle | theta X | theta Y | theta Z |
|---|---|---|---|---|
| straight | 1 → 2 | 0 | 0 | 0 |
| straight | 2 → 3 | −0.9 | 2.8 | 5.4 |
| straight | 3 → 4 | −0.1 | −2.4 | −2.8 |
| straight | 4 → 5 | −3.5 | −0.7 | −1.2 |
| straight | 5 → 6 | −2.3 | 3 | −2.6 |
| straight | 6 → stylocone | 12.2 | 1.8 | 1.3 |
| lateral | 1 → 2 | 0 | −13.3 | 0 |
| lateral | 2 → 3 | −0.03 | −6 | 5.4 |
| lateral | 3 → 4 | −1.3 | −25 | −3.1 |
| lateral | 4 → 5 | −3.6 | −6.6 | −1.2 |
| lateral | 5 → 6 | −2.4 | 0.3 | −2.5 |
| lateral | 6 → stylocone | 12.3 | −2 | 1.3 |
| flat side | 1 → 2 | 0 | 0 | 13 |
| flat side | 2 → 3 | −0.9 | 2.8 | 14.5 |
| flat side | 3 → 4 | −0.1 | −2.8 | 13.2 |
| flat side | 4 → 5 | −3.5 | 0.6 | 8.5 |
| flat side | 5 → 6 | −2.4 | 0.3 | −2 |
| flat side | 6 → stylocone | 12.2 | 1.8 | 10.1 |
| convex side | 1 → 2 | 0 | 0 | −10 |
| convex side | 2 → 3 | −0.9 | 2.8 | −13.5 |
| convex side | 3 → 4 | −0.1 | −2.4 | −42.1 |
| convex side | 4 → 5 | −3.5 | −0.7 | −23 |
| convex side | 5 → 6 | −2.3 | 3 | −7.5 |
| convex side | 6 → stylocone | 12.2 | 1.8 | −6.2 |

## 3.2. Range of motion

The joint centre for each pair of inferolaterals falls on the midline between the two members of the pair, at the dorsoventral midpoint, and near the distalmost point along the anterior–posterior axis. The joint centre of the stylocone falls at the geometric centre of the sphere fitted to the shape of the depression on the proximal surface (figure 2). The maximum range of motion of the proximal aulacophore and stylocone was 102° on the dorsoventral axis on the convex side, 57° on the flat side, and 53° laterally (table 1 lists the range of motion of each segment). These values are relative to the neutral reference position of the aulacophore ('straight' figure 1b). The distal aulacophore, with flat articular surfaces between successive ossicles, is notable for its lack of joint interfaces, which are fundamental for performing joint rotational motions and transmitting active forces to drive musculoskeletal-based locomotion.

## 4. Discussion

These results provide a new framework to test the three previously hypothesized strategies for aulacophore-powered locomotion in cornute stylophorans: via dorsoventral motion [13] or lateral

motion driven by a muscular distal aulacophore [14], or a muscular proximal aulacophore [14]. Although the aulacophore in *Phyllocystis crassimarginata* was theoretically capable of dorsoventral motion, it could not have performed the dorsoventral oscillations envisaged for aulacophore-first epifaunal locomotion in cornutes [13]. Based on the range of motion, the aulacophore was not flexible enough to be inserted into sediment at a relatively shallow angle and removed nearly perpendicular to the theca after sweeping through the substrate. It is clear that the aulacophore in this specimen of *P. crassimarginata* was not adapted for epifaunal locomotion using exclusively dorsoventral oscillations.

The cornute *Procothurnocystis owensi* was previously interpreted as using lateral oscillations of a muscular distal aulacophore to pull itself along [14: fig. 11]. Specimens of cornutes show various degrees of flexion of the distal aulacophore [14]. Thus, despite the planar surfaces of successive ossicles, the distal aulacophore of *P. crassimarginata* may also have been somewhat flexible. However, the flat morphology of the articular surfaces and lack of distinct joint interfaces indicate that the distal aulacophore would not have been able to exert contractile forces in a manner analogous to skeletal structures that incorporate joint rotational motions for locomotion in extant animals. Both proximal and distal surfaces of the ossicles in the distal aulacophore of the cornute *Ceratocystis* sp. were comprised of medium galleried stereom indicating the insertion of ligamentous collagenous tissue rather than muscle [12]. This suggests that the distal aulacophore was a semi-rigid structure that afforded an elastic response to the forces applied by the proximal aulacophore.

Woods & Jefferies [14] performed simulations with a wooden model based on specimens of the cornute *Cothurnocystis elizae* in which lateral motion was generated by the proximal aulacophore. They hypothesized that lateral movements of the aulacophore against the substrate moved the organism epifaunally in the direction of the arm. The angle of flexion necessary for this kind of locomotion [14: figs. 7,9] falls within our calculated range of motion, and the cornute *Ceratocystis* sp. provides evidence of musculature in the proximal aulacophore [12]. Lateral movement of the proximal aulacophore (coupled with dorsoventral motions of the proximal aulacophore in the power and recovery strokes) may have been used for locomotion along the surface of the substrate.

The mobility estimates for the specimen of *Phyllocystis crassimarginata* that we investigated represent a theoretical upper limit for the range of motion of the proximal aulacophore *in vivo* for this specimen. Measurements based on defleshed skeletons tend to overestimate mobility compared with estimates incorporating soft tissue [24,25]. Range of motion may also have been lower *in vivo* due to the presence of the tectals. Some morphological features of *Phyllocystis* disadvantaged it mechanically compared with many other stylophorans (i.e. asymmetry of the theca [26], absence of joint interfaces in the distal aulacophore), providing a somewhat low-bound for the locomotory capabilities of the entire clade.

Living echinoderms move to escape from predators, for example, or position the individual for feeding (including suspension feeding). Locomotory appendages in crinoids, asteroids and ophiuroids consist of serially repeated ossicles. The semi-rigid and probably non-muscular nature of the distal aulacophore in *P. crassimarginata* would have prevented crawling using motions similar to those of modern ophiuroid arms, for example, which show increased flexibility distally [16]. Nevertheless, *P. crassimarginata* could have used the aulacophore for aulacophore-first propulsion in the course of feeding or predator avoidance. The organism would also have been able to orient the aulacophore in the water column for feeding.

Past interpretations of aulacophore function have framed a role in locomotion and feeding as contrasting hypotheses [8,9,12,13,15]. However, there is no basis for considering these functions as mutually exclusive: appendages in extant echinoderms perform both feeding and locomotion (i.e. the arms of asteroids, brittle stars and crinoids). There is evidence of the presence of both musculature/connective tissue and water vascular tissue within the stylophoran aulacophore [9,12] strongly indicating that the arm supported both functions [27,28].

Applying our methodology to a range of disparate stylophoran taxa is critical to reveal the distribution of motion capabilities throughout the total group. The distal aulacophore in specimens of the mitrate *Rhenocystis latipedunculata* associated with locomotory trackways [8], for example, shows greater flexibility than that of *P. crassimarginata*, representing an interesting subject for future analysis. In addition, these trace fossils suggest that *R. latipedunculata* [8] moved with the aulacophore anterior; it is unclear whether this direction of motion was plausible for all stylophorans. Further analysis should also consider the influence of the cover plates and the tectals in effecting or inhibiting motion. Incorporating force application capabilities into our skeletal model (e.g. [17]) or creating a robotic model (e.g. [29]) would assist in addressing questions such as the most likely direction of motion as well as evaluating different hypotheses for aulacophore movement. The methodology presented here

can be applied to calculate range of motion in other fossil invertebrates with a rigid skeleton, including arthropods, and could be used to test broader hypotheses about evolutionary functional transitions in multiple taxa.

Ethics. We used no live animals in this study. The fossil specimen is housed by the Université Claude Bernard Lyon 1, Faculté des Sciences de Lyon (UCBL-FSL) and was not collected as part of this investigation.

Data accessibility. The micro-CT scan data and the 3D Maya model are available through the Dryad Digital Repository: https://dx.doi.org/10.5061/dryad.3sv28v7 [22].

Authors' contributions. E.G.C., J.R.H. and D.E.G.B. were involved in conceptualization and funding acquisition. E.G.C. processed the micro-CT data. E.G.C., J.R.H. and P.J.B. constructed the range of motion models. E.G.C., J.R.H. and D.E.G.B. interpreted the results and prepared the manuscript with input from P.J.B. All authors gave final approval for publication.

Competing interests. We declare we have no competing interests.

Funding. This project was funded by the National Science Foundation (NSF Award no. 1701830), the Yale Institute for Biospheric Studies and the Paleontological Society.

Acknowledgements. We thank Bertrand Lefebvre (Université Claude Bernard Lyon 1) for access to specimens and valuable discussion, and Bhart-Anjan Bhullar (Yale University), Ben Smith, Richard Bomphrey, Chris Richards, Eva Herbst, Krijn Michel, Andrew Cuff and Louise Kermode (Royal Veterinary College) for advice and assistance. We thank the reviewers, including Bertrand Lefebvre, Imran Rahman (Oxford University), Sébastien Clausen (Université des Sciences et Technologies de Lille), and the editors Emily Standen (University of Ottawa) and Kevin Padian (University of California, Berkeley) for their detailed and helpful comments.

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
