## [Reviewer comments · Royal Society Open Science]

Review History

RSOS-200191.R0 (Original submission)

Review form: Reviewer 1 (Trina Du)

Is the manuscript scientifically sound in its present form?

Yes

Are the interpretations and conclusions justified by the results?

Yes

Is the language acceptable?

Yes

Do you have any ethical concerns with this paper?

No

Have you any concerns about statistical analyses in this paper?

No

Recommendation?

Major revision is needed (please make suggestions in comments)

Comments to the Author(s)

See attached file (Appendix A).

Decision letter (RSOS-200191.R0)

10-Mar-2020

Dear Dr Clark,

The editors assigned to your paper ("Arm waving in stylophoran echinoderms: 3D mobility analysis illuminates cornute locomotion") have now received comments from reviewers. We would like you to revise your paper in accordance with the referee and Associate Editor suggestions which can be found below (not including confidential reports to the Editor). Please note this decision does not guarantee eventual acceptance.

Please submit a copy of your revised paper before 02-Apr-2020. Please note that the revision deadline will expire at 00.00am on this date. If we do not hear from you within this time then it will be assumed that the paper has been withdrawn. In exceptional circumstances, extensions may be possible if agreed with the Editorial Office in advance. We do not allow multiple rounds of revision so we urge you to make every effort to fully address all of the comments at this stage. If deemed necessary by the Editors, your manuscript will be sent back to one or more of the original reviewers for assessment. If the original reviewers are not available, we may invite new reviewers.

- Data accessibility

It is a condition of publication that all supporting data are made available either as supplementary information or preferably in a suitable permanent repository. The data accessibility section should state where the article's supporting data can be accessed. This section should also include details, where possible of where to access other relevant research materials such as statistical tools, protocols, software etc can be accessed. If the data have been deposited in an external repository this section should list the database, accession number and link to the DOI

for all data from the article that have been made publicly available. Data sets that have been deposited in an external repository and have a DOI should also be appropriately cited in the manuscript and included in the reference list.

<http://datadryad.org/submit?journalID=RSOS&manu=RSOS-200191>

- **Competing interests**

- **Authors' contributions**

- **Acknowledgements**

- **Funding statement**

Kind regards,

Andrew Dunn

on behalf of Professor Emily Standen (Associate Editor) and Kevin Padian (Subject Editor)

Associate Editor's comments (Professor Emily Standen):

Dear Dr. Elizabeth Clark,

We have now received sufficient review of your manuscript entitled Arm waving in stylophoran echinoderms: 3D mobility analysis illuminates cornute locomotion. The reviewer gives a very

positive report on your work along with several very clear and helpful suggestions on how to clarify and improve the impact of the manuscript itself.

In particular, clarification of clear hypotheses and predictions that are then tested by your model would greatly improve the papers contribution to addressing the main goal of the study. In addition, further information on your own data (please see reviewer comments) as well as an expanded introduction and discussion to fully address the impact of your data would greatly improve the manuscript's impact.

We look forward to seeing this manuscript re-submitted after a thorough address of the reviewer comments.

Sincerely,
Emily Standen

Comments to Author:

Reviewers' Comments to Author:
Reviewer: 1

Comments to the Author(s)
See attached file.

Author's Response to Decision Letter for (RSOS-200191.R0)

See Appendix B.

Decision letter (RSOS-200191.R1)

28-Apr-2020

Dear Dr Clark:

On behalf of the Editors, I am pleased to inform you that your Manuscript RSOS-200191.R1 entitled "Arm waving in stylophoran echinoderms: 3D mobility analysis illuminates cornute locomotion" has been accepted for publication in Royal Society Open Science subject to minor revision in accordance with the referee suggestions. Please find the referees' comments at the end of this email.

The reviewers and Subject Editor have recommended publication, but also suggest some minor revisions to your manuscript. Therefore, I invite you to respond to the comments and revise your manuscript.

- Ethics statement

- Data accessibility

<http://datadryad.org/submit?journalID=RSOS&manu=RSOS-200191.R1>

- Competing interests

- Authors' contributions

- Acknowledgements

- Funding statement

Because the schedule for publication is very tight, it is a condition of publication that you submit the revised version of your manuscript before 07-May-2020. Please note that the revision deadline will expire at 00.00am on this date. If you do not think you will be able to meet this date please let me know immediately.

To revise your manuscript, log into <https://mc.manuscriptcentral.com/rsos> and enter your Author Centre, where you will find your manuscript title listed under "Manuscripts with Decisions". Under "Actions," click on "Create a Revision." You will be unable to make your

revisions on the originally submitted version of the manuscript. Instead, revise your manuscript and upload a new version through your Author Centre.

on behalf of Professor Emily Standen (Associate Editor) and Kevin Padian (Subject Editor)
openscience@royalsociety.org

Author's Response to Decision Letter for (RSOS-200191.R1)

See Appendix C.

Decision letter (RSOS-200191.R2)

12-May-2020

Dear Dr Clark,

It is a pleasure to accept your manuscript entitled "Arm waving in stylophoran echinoderms: 3D mobility analysis illuminates cornute locomotion" in its current form for publication in Royal Society Open Science.

on behalf of Professor Emily Standen (Associate Editor) and Kevin Padian (Subject Editor)
openscience@royalsociety.org

Appendix A

This paper addresses an interesting issue and has the potential to make a substantial contribution to knowledge of this enigmatic taxon. Overall, I found the method appropriate and well described, and the paper well written. My main comment is that this paper would benefit from more content, particularly more comprehensive details on the ROMs of individual elements. Much of this additional information may already be collected by the authors, or is collectable from the same 3D models.

MAJOR COMMENTS

1. (Line 56) “Several hypotheses for aulacophore range of motion and locomotion strategies have been proposed [5-7, 11, 13-14].”

-I would appreciate a description of these hypothesized ROM and locomotion strategies here in the introduction, in addition to in the discussion. As stated in the final paragraph of the introduction, the objective of this study is to use the 3D model to test the plausibility of these hypotheses. Specifying exactly what hypotheses are being compared, especially their predictions about range of motion, would be helpful.

2. (Line 150)

- What exactly are these angles? I presume from looking at Fig. 1 that they are the angle formed between the tip of the stylocone in the straight and maximally flexed positions, but this is unclear in the text.

General

3. Line 117 states that the ROM of each inter-ossicle joint was measured. It would be nice to see these presented in the results section along with the total angle of the aulacophore, especially since it appears from Fig. 1 that the degree of bending is variable along the length of the structure. I would also like to see the ROM of the proximal aulacophore-stylocone joint reported, since it seems to have a distinct shape and features prominently in Fig. S1, but doesn't seem to bend at all in Fig. 1. Does this rigidity between the proximal aulacophore-stylocone have any functional consequences?

4. I am concerned about the ROM result presented for the dorsolateral axis of the convex side, given that the tectals were not included in the analysis. Wouldn't the lack of tectals explain why the ROM on the convex side is nearly double that of the flat side? If this is the case, I am unsure of the value in reporting a 102° ROM that is based on an artefact from specimen preservation. Although mentioned on line 192 and 214, I think this topic requires greater commentary.

5. It would be worthwhile to also model the ROM of the distal aulacophore. Although the authors provide justification for why it is not included (probably inflexible, lack of muscle attachments), if the motivation of this study was to generate an objective measurement of theoretical maximal ROM without biases from descriptive morphology, shouldn't the distal aulacophore be included? Furthermore, if potential effects of soft tissues are not considered for the ROM of the proximal aulacophore skeleton, why should their presence/absence preclude an analysis of the distal skeleton?

MINOR COMMENTS

6. (Line 74) “Several morphological features of *Phyllocystis* are mechanically disadvantageous in comparison to many other stylophorans...”

-Given that the rest of the paragraph establishes the objective and conceptual approach of the study, this seems like an odd way to end the introduction. Perhaps this subject of *Phyllocystis*-specific traits would be better placed in the discussion.

7. (Line 132) “The width of the pair of inferolaterals increases distally to accommodate the succeeding segment...”

-It looks from Fig. 1 that the width decreases distally (distal segments are nested inside proximal ones?).

8. (Line 204) “There is evidence of the presence of both musculature/connective tissue and water vascular tissue within the stylophoran aulacophore [9,12]...”

-Some clarification is needed here as the cited papers seem to support either the presence of muscle or the presence of a water vascular system, not both simultaneously (9 concludes that it is a water vascular system not a muscular stem, 12 concludes that is a muscular stem with no evidence of tube feet).

General

9. I don't think it's stated in the manuscript whether the convex/flat side is the upper/lower surface of the organism. Is this important?

10. It would be worthwhile to turn Fig. S1 into a main figure, particularly since there are no close-up images of the stylocone in Fig. 1.

Appendix B

Response to Reviewer Comments: RSBL-2019-0872

We are grateful for the reviewer's and handling editor's prompt and detailed comments on our manuscript. We have incorporated all of the suggestions provided into a revised version of the manuscript, with a few exceptions. Line numbers refer to the revised word document. Text that has been added to the manuscript is indicated in red.

Reviewer 1 writes that this paper “has the potential to make a substantial contribution to knowledge of this enigmatic taxon.” They primarily suggested adding additional detail about the scientific context in the introduction and details about the range of motion measurements in the results.

MAJOR COMMENTS

1. Line 56: “Several hypotheses for aulacophore range of motion and locomotion strategies have been proposed [5-7, 11, 13-14].” I would appreciate a description of these hypothesized ROM and locomotion strategies here in the introduction, in addition to in the discussion. As stated in the final paragraph of the introduction, the objective of this study is to use the 3D model to test the plausibility of these hypotheses. Specifying exactly what hypotheses are being compared, especially their predictions about range of motion, would be helpful.

Changes made: Text detailing the hypotheses has been added to the introduction (line 58-72).

2. Line 150: What exactly are these angles? I presume from looking at Fig. 1 that they are the angle formed between the tip of the stylocone in the straight and maximally flexed positions, but this is unclear in the text.

Changes made: These values refer to the angle of flexion relative to the neutral reference pose of the aulacophore (“straight,” in figure 1B). We have added text to clarify this (line 165-166).

General

3. Line 117 states that the ROM of each inter-ossicle joint was measured. It would be nice to see these presented in the results section along with the total angle of the aulacophore, especially since it appears from Fig. 1 that the degree of bending is variable along the length of the structure. I would also like to see the ROM of the proximal aulacophore-stylocone joint reported, since it seems to have a distinct shape and features prominently in Fig. S1, but doesn't seem to bend at all in Fig. 1. Does this rigidity between the proximal aulacophore-stylocone have any functional consequences?

Changes made: The range of motion at the joint created by the stylocone and the proximal aulacophore tends to be relatively lower than those between the segments of the proximal aulacophore; however, as the stylocone is at the end of the kinematic chain created by the segments of the proximal aulacophore, the movement capability of that structure is the integrated range of the proximal segments. The stylocone has been known prior as a structure serving to

connect the disparate proximal and distal aulacophore sections; this interpretation is supported by the relatively low mobility observed. We have added these angle measurements as a supplementary table referenced in the results section (lines 164-165).

4. I am concerned about the ROM result presented for the dorsolateral axis of the convex side, given that the tectals were not included in the analysis. Wouldn't the lack of tectals explain why the ROM on the convex side is nearly double that of the flat side? If this is the case, I am unsure of the value in reporting a 102° ROM that is based on an artefact from specimen preservation. Although mentioned on line 192 and 214, I think this topic requires greater commentary.

Changes not made (no changes necessary): We agree with the reviewer that the tectals have the potential to influence the overall range of motion of the aulacophore, and it is unfortunate that they are not preserved in articulation in this specimen. Whether or not the tectals would have limited the range of motion is entirely dependent on their position relative to the inferolaterals. This information is unfortunately not available for this specimen, and we decided it was best not to speculate about the implications of this absent information. We address the limitations of this specimen (line 111). We remark that future analyses of specimens with articulating tectals preserved may resolve this issue (lines 237-239), and that the actual range of motion may have been lower because of this (lines 209-210). We believe this sufficiently addresses the reviewer's concerns and provides direction for future research.

5. It would be worthwhile to also model the ROM of the distal aulacophore. Although the authors provide justification for why it is not included (probably inflexible, lack of muscle attachments), if the motivation of this study was to generate an objective measurement of theoretical maximal ROM without biases from descriptive morphology, shouldn't the distal aulacophore be included? Furthermore, if potential effects of soft tissues are not considered for the ROM of the proximal aulacophore skeleton, why should their presence/absence preclude an analysis of the distal skeleton?

Changes made: Because the articulated surfaces in the distal aulacophore are flat and lack a distinct joint interface, we cannot assume that these ossicles functioned in the same way as any other structures in the animal kingdom known to us. They are certainly unlike typical structures used for locomotion in extant animals (that employ hinge joints, ball and socket joints, etc.). Text has been added to this effect in the results section (lines 166-169). The discussion of the presence of collagenous tissue and lack of musculature in the space between the ossicles (based on [12]) provides the available evidence regarding the nature of this unusual structure. The lack of musculature suggests that these connections were relatively rigid or, at the very least, that musculoskeletal-based motions of the distal aulacophore were not used to perform locomotion (lines 187-190).

The proximal aulacophore has obvious interfaces between successive elements, making it possible to calculate range of motion. The available evidence for the architecture of the soft tissue within the proximal aulacophore is described in the text (lines 201-202).

MINOR COMMENTS

6. Line 74: “Several morphological features of Phyllocystis are mechanically disadvantageous in comparison to many other stylophorans...” Given that the rest of the paragraph establishes the objective and conceptual approach of the study, this seems like an odd way to end the introduction. Perhaps this subject of Phyllocystis-specific traits would be better placed in the discussion.

Changes made: We have moved this text to the discussion as per the reviewer’s suggestion (lines 210-214).

7. Line 132: “The width of the pair of inferolaterals increases distally to accommodate the succeeding segment...” It looks from Fig. 1 that the width decreases distally (distal segments are nested inside proximal ones?).

Changes made: The width increases along the length of each individual ossicle distally to accommodate the next successive segment; the overall width of each sequential segment decreases distally. We have changed the text to clarify this (lines 144-146).

8. Line 204: “There is evidence of the presence of both musculature/connective tissue and water vascular tissue within the stylophoran aulacophore [9,12]...” Some clarification is needed here as the cited papers seem to support either the presence of muscle or the presence of a water vascular system, not both simultaneously (9 concludes that it is a water vascular system not a muscular stem, 12 concludes that is a muscular stem with no evidence of tube feet).

Changes made: As the reviewer notes, reports in the past have asserted a feeding and locomotory role for the aulacophore as contrasting hypotheses. However, there is no justification for this dichotomy, as there are many examples of structures in extant echinoderms that perform both. There is also compelling evidence for both interpretations, and no logical reason for why these two functions would be mutually exclusive. We have clarified this in lines 224-230.

General

9. I don’t think it’s stated in the manuscript whether the convex/flat side is the upper/lower surface of the organism. Is this important?

Changes not made (no change necessary): We have intentionally chosen our terminology as stylophorans are one of the only animals for which a consensus regarding the identity of the dorsal and ventral surfaces has yet to emerge, and contributing to this debate is outside the scope of this paper.

10. It would be worthwhile to turn Fig. S1 into a main figure, particularly since there are no close-up images of the stylocone in Fig. 1.

Changes made: We have changed Fig. S1 to Fig. 2 in the main text. We have updated the text accordingly (lines 128, 162, 341-347).

Appendix C

Yale University

Mailing address (U.S. Postal Service)
Department of Geology and Geophysics
Kline Geology Laboratory
P.O. Box 208109
New Haven
Connecticut 06520-8109, U.S.A.

Campus address (UPS and FedEx)
Kline Geology Laboratory
210 Whitney Avenue
New Haven
Connecticut 06511
Telephone/Fax: 203 432-8590

30th April, 2020

Dear Editor (*Royal Society Open Science*),

We are pleased that our manuscript RSOS-20019 entitled “Arm waving in stylophoran echinoderms: 3D mobility analysis illuminates cornute locomotion” has been accepted for publication subject to minor revisions. We have incorporated these suggestions into the final version of the manuscript. Per the reviewer’s suggestions, we now include two figures and a table to be included in the main text. The micro-CT data and digital models are accessible via Dryad.

Yours sincerely,

Elizabeth G. Clark
Postdoctoral Fellow, School of the Environment
Yale University

Derek E.G. Briggs
G. Evelyn Hutchinson Professor of Geology and Geophysics
Curator of Invertebrate Paleontology, Yale Peabody Museum

John R. Hutchinson
Professor of Evolutionary Biomechanics
Royal Veterinary College, UK

Peter J. Bishop
Postdoctoral Researcher in Evolutionary Biomechanics
Royal Veterinary College, UK

Response to Reviewer Comments: RSBL-2019-0872.R1

The Editors were of the view that the two figures and single table in the supplementary materials could, with proper formatting, be included in the main body of the paper. They wanted to stress that the paper is otherwise ready for acceptance, but they were a little concerned that data of this type can become a little detached from the original paper (though the ESM are linked to the published paper) and may be harder to find for readers.

Changes made: We now include these two figures and the table into the main body of the paper.